# Diverse signatures of convergent evolution in cactus-associated yeasts

Carla Gonçalves[1,2,3,4], Marie-Claire Harrison[1,2], Jacob L. Steenwyk[1,2,5], Dana A. Opulente[6,7], Abigail L. LaBella[1,2,8], John F. Wolters[6], Xiaofan Zhou[1,2,9], Xing-Xing Shen[1,2,10], Marizeth Groenewald[11], Chris Todd Hittinger[6], Antonis Rokas[1,2]*

1 Department of Biological Sciences, Vanderbilt University, Nashville, Tennessee, United States of America, 2 Evolutionary Studies Initiative, Vanderbilt University, Nashville, Tennessee, United States of America, 3 Associate Laboratory i4HB—Institute for Health and Bioeconomy and UCIBIO—Applied Molecular Biosciences Unit, Department of Life Sciences, NOVA School of Science and Technology, Universidade NOVA de Lisboa, Caparica, Portugal, 4 UCIBIO-i4HB, Departamento de Ciências da Vida, Faculdade de Ciências e Tecnologia, Universidade Nova de Lisboa, Caparica, Portugal, 5 Howards Hughes Medical Institute and the Department of Molecular and Cell Biology, University of California, Berkeley, Berkeley, California, United States of America, 6 Laboratory of Genetics, DOE Great Lakes Bioenergy Research Center, Center for Genomic Science Innovation, J. F. Crow Institute for the Study of Evolution, Wisconsin Energy Institute, University of Wisconsin-Madison, Madison, Wisconsin, United States of America, 7 Biology Department, Villanova University, Villanova, Pennsylvania, United States of America, 8 Department of Bioinformatics and Genomics, University of North Carolina at Charlotte, Charlotte, North Carolina, United States of America, 9 Guangdong Province Key Laboratory of Microbial Signals and Disease Control, Integrative Microbiology Research Center, South China Agricultural University, Guangzhou, China, 10 College of Agriculture and Biotechnology and Centre for Evolutionary & Organismal Biology, Zhejiang University, Hangzhou, China, 11 Westerdijk Fungal Biodiversity Institute, Utrecht, the Netherlands

* antonis.rokas@vanderbilt.edu

**Data Availability Statement:** All input data for all analyses and raw results files can be found in

## Abstract

Many distantly related organisms have convergently evolved traits and lifestyles that enable them to live in similar ecological environments. However, the extent of phenotypic convergence evolving through the same or distinct genetic trajectories remains an open question. Here, we leverage a comprehensive dataset of genomic and phenotypic data from 1,049 yeast species in the subphylum Saccharomycotina (Kingdom Fungi, Phylum Ascomycota) to explore signatures of convergent evolution in cactophilic yeasts, ecological specialists associated with cacti. We inferred that the ecological association of yeasts with cacti arose independently approximately 17 times. Using a machine learning–based approach, we further found that cactophily can be predicted with 76% accuracy from both functional genomic and phenotypic data. The most informative feature for predicting cactophily was thermotolerance, which we found to be likely associated with altered evolutionary rates of genes impacting the cell envelope in several cactophilic lineages. We also identified horizontal gene transfer and duplication events of plant cell wall–degrading enzymes in distantly related cactophilic clades, suggesting that putatively adaptive traits evolved independently through disparate molecular mechanisms. Notably, we found that multiple cactophilic species and their close relatives have been reported as emerging human opportunistic pathogens, suggesting that the cactophilic lifestyle—and perhaps more generally lifestyles favoring thermotolerance—might preadapt yeasts to cause human disease. This work underscores the potential of a multifaceted approach involving high-throughput genomic

Figshare: https://doi.org/10.6084/m9.figshare.
24114381.

**Funding:** This work was supported by the National
Science Foundation (Grants DEB-2110403 to CTH
and DEB-2110404 to AR). Research was also
supported by DOE Great Lakes Bioenergy Research
Center, funded by BER Office of Science (Grant DE-
SC0018409 to CTH), USDA National Institute of
Food and Agriculture (Hatch Projects 1020204 and
7005101, to CTH), an H. I. Romnes Faculty
Fellowship, supported by the Office of the Vice
Chancellor for Research and Graduate Education
with funding from the Wisconsin Alumni Research
Foundation (to CTH). Research was also supported
by the National Institutes of Health/National
Institute of Allergy and Infectious Diseases Grant
(R01AI153356 to AR) and the Burroughs
Wellcome Fund (to AR). Research was also
supported by the National Key R&D Program of
China (Grant 2022YFD1401600 to XXS), the
National Science Foundation for Distinguished
Young Scholars of Zhejiang Province (Grant
LR23C140001 to XXS), and the Fundamental
Research Funds for the Central Universities (Grant
226-2023-00021 to XXS). Research was partially
supported by the National Institutes of Health
(Grant T32HG002760-16 to JFW) and a National
Science Foundation Grant Postdoctoral Research
Fellowship in Biology (1907278 to JFW). Research
was also supported by Fundação para a Ciência e a
Tecnologia, in the scope of the project UIDP/
04378/2020 and UIDB/04378/2020 of the
Research Unit on Applied Molecular Biosciences -
UCIBIO and the project LA/P/0140/2020 of the
Associate Laboratory Institute for Health and
Bioeconomy - i4HB, and grants PTDC/BIA-EVL/
0604/2021 (to CG) and the Federation of European
Microbiological Societies (FEMS grant FEMS-GO-
2019-537 to CG). JLS is a Howard Hughes Medical
Institute Awardee of the Life Sciences Research
Foundation. The funders had no role in study
design, data collection and analysis, decision to
publish, or preparation of the manuscript.

**Competing interests:** JLS is a scientific advisor for
WittGen Biotechnologies. JLS is an advisor for
ForensisGroup Inc. AR is a scientific consultant for
LifeMine Therapeutics, Inc. All other authors have
declared that no competing interests exist.

**Abbreviations:** BH, Benjamini–Hochberg; GO, gene
ontology; gw-RSCU, gene-wise relative
synonymous codon usage; HGT, horizontal gene
transfer; LDT, Lipomycetales/Dipodascales/
Trigonopsidales; LRT, likelihood ratio test; MRCA,
most recent common ancestor; OG, orthologous
group; RER, relative evolutionary rate; RF, random

and phenotypic data to shed light onto ecological adaptation and highlights how convergent
evolution to wild environments could facilitate the transition to human pathogenicity.

## Introduction

Convergent evolution, the repeated evolution of similar traits among distantly related taxa, is
ubiquitous in nature and has been documented across all domains of life [1–3]. Convergence
typically arises when organisms occupy similar ecological niches or encounter similar condi-
tions and selective pressures; facing similar selective pressures, organisms from distinct line-
ages often evolve similar adaptations.

Independently evolved phenotypes often share the same genetic underpinnings (parallel
molecular evolution) [4–7], such as similar mutations in specific genes [4,5,8], but can also
arise through distinct molecular paths and by distinct evolutionary mechanisms [9,10], such as
gene duplications [11–13], gene losses [14–16], or horizontal gene transfer (HGT) events
[17,18]. Molecular signatures of convergence can also be inferred from independent shifts in
overall evolutionary rates [19] and examined at higher hierarchical levels of molecular organi-
zation, such as functions or pathways [20]. For instance, comparing rates of evolution across
distantly related animal lineages could pinpoint convergent slowly evolving genes involved in
adaptive functions [21] or convergent rapidly evolving genes indicating parallel relaxed con-
straints acting on dispensable functions [22]. Parallel molecular changes are common across
all domains of life [9], but their occurrence can be reduced by mutational epistasis or the poly-
genic nature of some phenotypic traits [10,23,24], particularly when studying convergence in
distantly related organisms.

Fungi exhibit very high levels of evolutionary sequence divergence [25]; the amino acid
sequence divergence between the baker's yeast *Saccharomyces cerevisiae* and the human com-
mensal and opportunistic pathogen *Candida albicans*, both members of subphylum Saccharo-
mycotina (one of the 3 subphyla in Ascomycota, which is one of the more than 1 dozen fungal
phyla), is comparable to the divergence between humans and sponges [26]. Due to their very
diverse genetic makeups, convergent phenotypes arising in fungi might involve distinct genetic
determinants and/or mechanisms, including HGT [27,28], a far less common mechanism
among animals (but see [29,30]).

Saccharomycotina yeasts are ecologically diverse, occupy diverse ecosystems [31], and vary
considerably in their degree of ecological specialization ranging from cosmopolitan generalists
to ecological specialists. For instance, *Sugiyamaella* yeasts are mostly isolated from insects [32]
and most *Tortispora* species have been almost exclusively found in association with cacti plants
[33]. The cactus environment accommodates numerous yeast species rarely found in other
niches [34–37]. Moreover, cactophilic yeasts are part of a model ecological system involving
the tripartite relationship between cactus, yeast, and *Drosophila* [34,36,38,39]. Cactophilic
yeasts use necrotic tissues of cacti as substrates for growth [40] while serving as a food source
to cactophilic *Drosophila*. Cactophilic flies (and other insects) play, in turn, a crucial role in the
yeast's life cycle by acting as vectors [34].

In *Drosophila*, the adoption of cacti as breeding and feeding sites evolved around 16 to 21
million years ago (Mya) and is considered one of the most extensive and successful ecological
transitions within the genus [41]. Cactophilic *Drosophila* thrive across a wide range of cacti
species that differ in the profiles of toxic metabolites they produce—*Opuntia* species, com-
monly called prickly pear cactus, generally contain fewer toxic metabolites than columnar

forest; SCO, single-copy ortholog; tAI, tRNA
adaptation index.

cacti species [40] and are likely the ancestral hosts [41]. The distinctive characteristics of the cacti environment [42] seemingly selected for adaptive traits across cactophilic *Drosophila*, such as high resistance to heat, desiccation, and toxic alkaloid compounds produced by certain types of cacti [43–48]. These traits are likely associated with several genomic signatures (for instance, positive selection, gene duplications, HGT) impacting multiple functions, such as water preservation or detoxification [45–50].

Contrasting with *Drosophila*, where cactophily is largely found within the monophyletic *D. repleta* group [41], molecular phylogenetic analyses revealed that cactophilic yeasts belong to phylogenetically distinct clades, indicating that association with cacti evolved multiple times, independently, in the Saccharomycotina [37]. While relevant ecological and physiological information of cactus-associated yeasts is available [35–37,39], the genetic changes that facilitated the convergent evolution of multiple yeast lineages to the cacti environment are unknown.

Benefiting from the wealth of genomic and phenotypic data available for nearly all known yeast species described in the subphylum Saccharomycotina [51] and cross-referencing with the ecological data available from the cactus-yeast-*Drosophila* system [35–37,52–60], we employed a high-throughput framework to detect signatures of convergent evolution in 17 independently evolved lineages of cactophilic yeasts. Using a machine learning algorithm, we uncovered distinctive phenotypic traits enriched among cactus-associated yeasts, including the ability to grow at high ($\geq$37°C) temperatures. We found that thermotolerance might be related to distinctive rates of evolution in functions impacting the integrity of the cell envelope, some of which are under positive selection in distantly related cactophilic clades. Gene family analyses identified gene duplication and HGT events involving plant cell wall–degrading enzymes in distinct clades, suggesting adaptations associated with feeding on plant material. These results reveal that convergence to cactophily by distinct lineages of Saccharomycotina yeasts was accomplished through diverse evolutionary mechanisms acting on distinct genes, although some of these are involved in similar biological functions. Interestingly, we found that several cactus-associated yeasts and close relatives have been reported as emerging opportunistic human pathogens raising the hypothesis that fungi inhabiting certain wild environments may be preadapted for opportunistic pathogenicity. More broadly, we advocate for a methodological framework that couples diverse lines of genomic, phenotypic, and ecological data with multiple analytical approaches to investigate the plurality of evolutionary mechanisms underlying ecological adaptation.

## Results

### Yeast cactophily likely evolved independently 17 times

We examined the ecological association of yeasts with the cacti environment across a dataset of 1,154 strains from 1,049 yeast species. Yeast-cacti associations vary substantially in their strengths [35]. Some cactus-associated yeast species are considered cosmopolitan, being commonly isolated from cacti but also other environments (henceforth referred to as transient), whereas others are strictly cactophilic, defined as those almost exclusively isolated from cacti (S1 Table; note that this classification is based on the available ecological information, which may be impacted by sampling bias and other sampling issues—it is possible that strictly cactophilic species could also be found in other, yet unsampled, environments). We observed that strictly cactophilic species are found across almost the entire Saccharomycotina subphylum spanning from the Trigonopsidales (i.e., *Tortispora* spp.) [33] to the Saccharomycetales (i.e., *Kluyveromyces starmeri*) [52] (Figs 1 and S1). Using the yeast phylogeny and distribution of

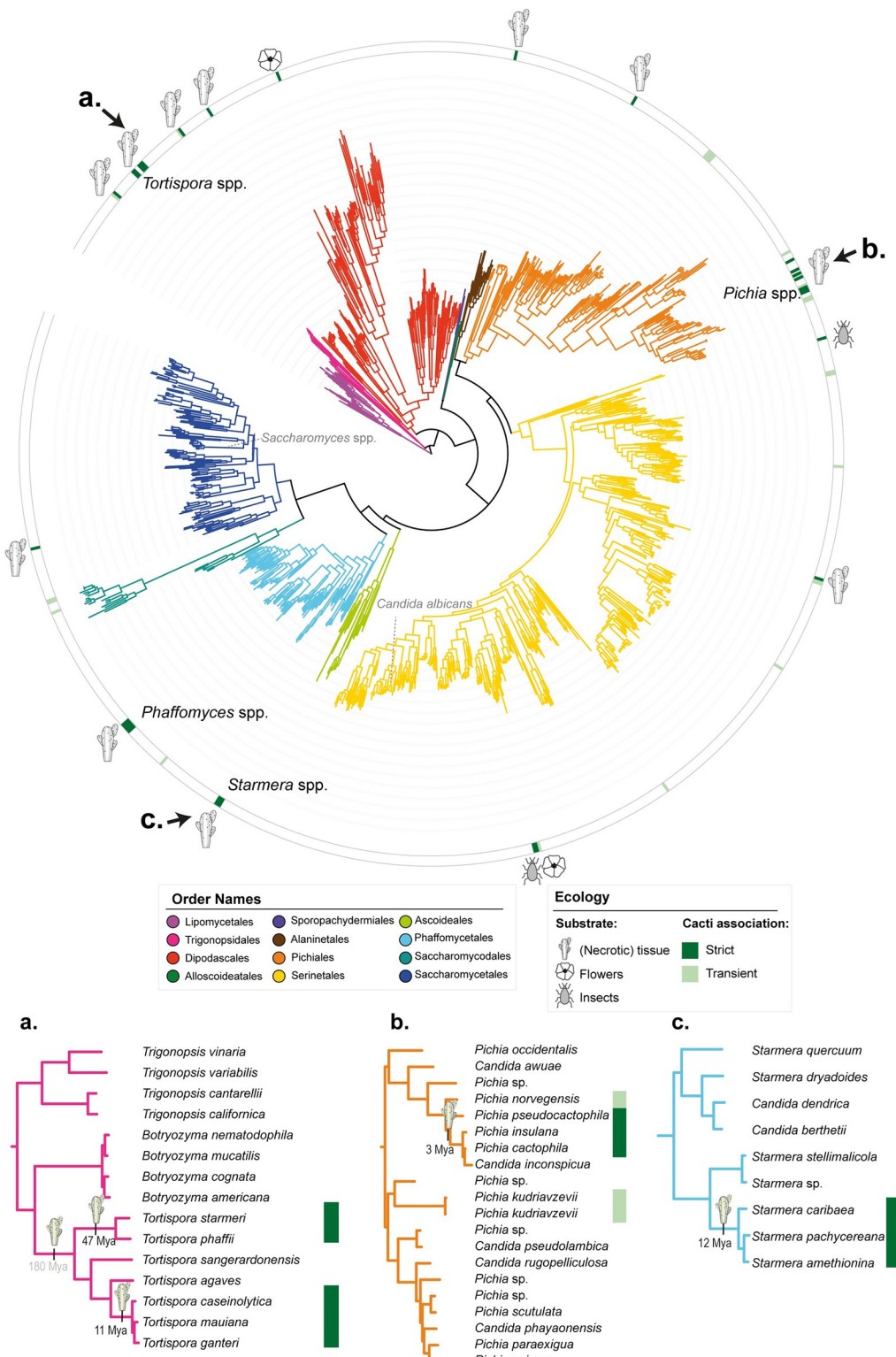

**Fig 1. Yeast cactophily originated repeatedly and at different times. (top)** Genome-wide–based phylogeny of the subphylum Saccharomycotina [51] depicting the different types of ecological association of strictly cactophilic yeast species with the cacti environment: (necrotic) cacti tissues, which include both *Opuntia* spp. and columnar cacti, cacti flowers, and cacti-visiting insects. Cactophilic lineages and well-known yeasts, such as *Saccharomyces* spp. and *Candida albicans*, are noted on the phylogeny. (**a, b,** and **c**) Subtrees of 3 cactophilic clades: (**a**) *Tortispora*, (**b**) *Starmera*, and (**c**)

*Pichia*. Estimated times of origin, as determined in [51], for the emergence of cactophily are represented for these 3 example clades. The data underlying this Figure can be found in https://doi.org/10.6084/m9.figshare.24114381.

(strict) cactophily in an ancestral state reconstruction, we inferred a total of 17 origins for the evolution of cactophily (S2 Fig).

Cactophily is found in single species belonging to different orders, but it also involves nearly an entire genus (i.e., *Tortispora*). Specifically, 7 of the 17 instances of cactophily evolution involve clades containing 2 or more species while the remaining 10 involve single species, suggesting that different taxa evolved this ecological association at different times (Figs 1 and S1). For instance, cross-referencing relaxed molecular clock analyses of the yeast phylogeny [51] with ancestral state reconstructions suggests that cactophily in *Tortispora* likely emerged twice, once in the most recent common ancestor (MRCA) of *T. starmeri*/*T. phaffi* around 47 Mya, and in the MRCA of *T. caseinolytica*/*T. mauiana*/*T. ganteri* around 11 Mya (Fig 1). An alternative hypothesis would place the emergence of cactophily in the MRCA of the *Tortispora* genus around 180 Mya; however, this hypothesis is inconsistent with the estimated origin of the Cactaceae family (35 Mya) [61]. Cactophily in the genus *Starmera* and in the *Pichia cactophila* clade emerged more recently, most likely around 12 and 3 Mya, respectively (Fig 1).

We also observed that cactus-associated yeasts seemingly exhibit significant niche partitioning (S1 Table). For example, *T. ganteri* is typically isolated from columnar cacti, while its close relative *T. caseinolytica* is more commonly found in *Opuntia* spp. [33]. Furthermore, *P. cactophila* is considered a generalist cactophilic yeast, being widely distributed across a wide range of cacti species [37], while closely related *P. heedii* has been predominantly found in association with certain species of columnar cacti [39]. However, many species (for instance, *T. starmeri* or *P. insulana*) alternate between the 2 types of cacti [33,58], similar to some *Drosophila* species [41]. Other strictly cactophilic species, such as *Wickerhamiella cacticola* or *Kodamaea nitidulidarum*, are associated with cacti flowers and/or flower-visiting insects, like beetles, and not with necrotic cacti tissues [59,60,62].

## Detecting signatures of convergent evolution in cactophilic yeasts

We envision 3 distinct scenarios that may capture how different yeast lineages convergently evolved cactophily (Fig 2A):

Scenario I: Convergent phenotypes and genotypes

Selective pressures associated with the cacti environment (for instance, high temperature, desiccation, or presence of toxic compounds) favor similar phenotypic traits that evolved through the same genomic mechanisms;

Scenario II: Convergent phenotypes through divergent genotypes

Selective pressures associated with the cacti environment favor similar phenotypic traits across cactophilic species, but different evolutionary mechanisms (for instance, gene duplication, HGT) and/or genes contribute to phenotypic convergence of different lineages. In this scenario, similar phenotypes emerge through distinct evolutionary trajectories;

Scenario III: Divergent phenotypes and genotypes

Distinct phenotypic landscapes are explored by distinct clades when thriving in the same environment (niche partitioning); different evolutionary mechanisms contribute to these phenotypes.

To explore which scenario(s) best reflect(s) the process of yeast adaptation to the cacti environment, we developed a framework for identifying signatures of adaptation and convergence from high-throughput genomic and phenotypic data [51] (Fig 2B). First, we employed a random forest (RF) classifier to identify phenotypic and genetic commonalities that distinguish

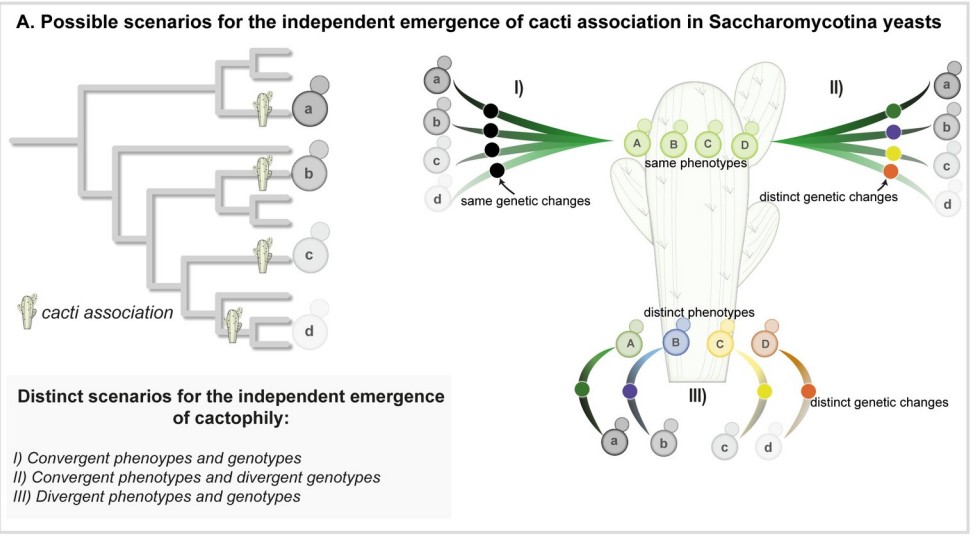

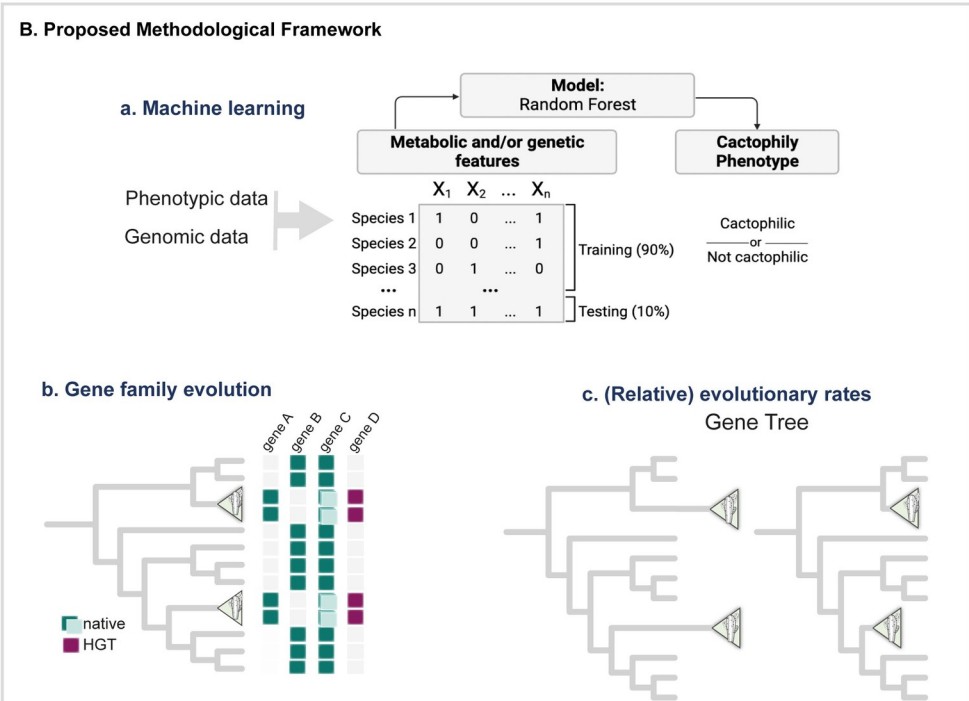

**Fig 2. Alternative scenarios for the evolution of yeast cactophily and methodological framework employed in this study.** (**A**) Methodological framework for investigating signatures of convergent evolution of yeast cactophily. Association with cacti evolved multiple times in yeasts from distinct genetic backgrounds, which are represented by distinct shades of gray. Convergent adaptation to the cacti environment might have involved convergence at both phenotypic and genomic levels (Scenario I), phenotypic convergence through distinct molecular paths (Scenario II), or lack of convergence at both phenotypic and genotypic levels (Scenario III). Each of these scenarios is tested using a methodological framework (**B**) involving (a) machine learning approach wherein a model is trained to distinguish cactophilic from non-cactophilic yeasts from genomic and phenotypic features; (b) gene family analyses to find evidence of gene duplications, losses, and HGT that might have occurred in cactophilic yeasts; and (c) evaluation of signatures of positive selection and of changes in relative evolutionary rates (evolving faster or slower) in branches leading to cactophilic clades.

cactophilic from non-cactophilic yeasts. We would expect that our RF classifier would yield highly accurate predictions for Scenario I, intermediate accuracy for Scenario II, and lack of accuracy for Scenario III. For instance, in Scenario I, where association with cacti would involve the evolution of similar phenotypes encoded by the same genomic paths, we expect that the accuracy of prediction obtained would be near 100% as both genomic and phenotypic features would be shared by all cactophilic species. In contrast, Scenario III, which implies that the distinct cactophilic lineages experienced distinct changes and display distinct phenotypes, we would expect that our accuracy of prediction would be close to random (i.e., 50%) because no genetic or phenotypic feature would be predictive of cactophily.

Second, we inspected patterns of gene presence/absence due to gene duplication, and HGT. Third, we investigated genome-wide evolutionary rates to detect signatures of convergence in evolutionary rates and of positive selection in individual genes (Fig 2B), which have been also frequently implicated in adaptive evolution [5,19,21,22,63]. Specifically, for the detection of convergent evolutionary rates, we adopted an approach that identifies genes with evolutionary rate (i.e., number of amino acid substitutions per site) shifts across a phylogeny including multiple cactophilic and non-cactophilic species and correlates these shifts with the independent emergence of cacti association [64]. The analyses of gene family and evolutionary rates allow us to identify which genomic features (genes) and mechanisms (duplication, HGT, or altered evolutionary rates) may be associated with the common phenotypic features identified in the machine learning analyses. Under Scenario I, we would expect to find similar mechanisms (positive selection, rapid/slow evolutionary rates, HGT, or duplication) impacting the same phenotype across cactophilic clades. Under Scenario II, we would expect to find distinct genetic mechanisms/genes affecting the same phenotypes across cactophilic clades. Under Scenario III, we would expect an absence of shared mechanisms, genes, and phenotypes across clades.

We applied this methodological framework to study convergence in ecological specialization in yeasts but note that it can be applied more generally to study the process of convergent or adaptive evolution.

## Specific metabolic and genomic traits predict cactophily

To investigate if cactophilic yeasts share similar phenotypic and/or genomic traits, we used a dataset of 1,154 yeast strains [51], from which 54 are either strictly cactophilic (rarely found in other environments, $n = 31$) or transient (frequently isolated from cacti but cosmopolitan, $n = 21$) (S1 Table). Functional genomic annotations (KEGG– 5,043 features) and physiological data (122 features) were retrieved from Opulente and colleagues [51] and used as features in a supervised RF classifier trained to distinguish cactophilic from non-cactophilic yeasts. By training 20 independent RF runs using randomly selected balanced datasets (54 non-cactophilic species randomized each time and the 54 cactophilic species), we correctly identified an average of 38 (approximately 70%) cactophilic species (Fig 3A), yielding an overall accuracy and precision of 72% and 73%, respectively. Species incorrectly assigned in 10 or more independent runs were equally distributed across strictly and transiently cactophilic groups (7 in each) (S2 Table). We next repeated the analysis considering only strictly cactophilic species (transient species were considered non-cactophilic) and obtained slightly higher accuracy (76%) and precision (76%) (Fig 3B). Notably, correct classifications were obtained across phylogenetically distantly related genera (for instance, *Tortispora*, *Phaffomyces*, or *Pichia*) (Fig 3A), while incorrect classifications were obtained for species belonging to cactophilic clades in which correct classifications were obtained; for instance, despite being closely related, we obtained both correct and incorrect classifications for cactophilic species belonging to the

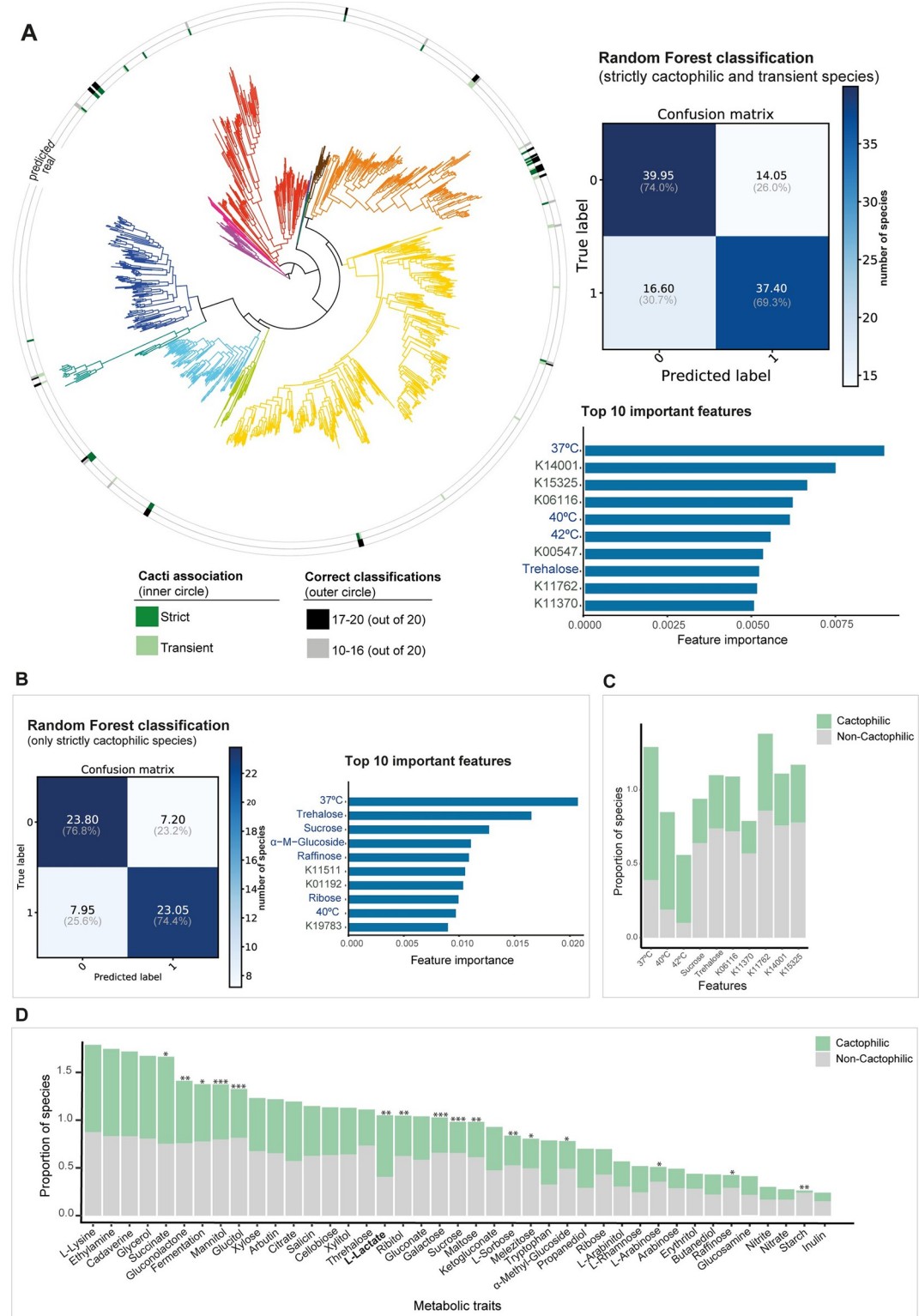

**Fig 3. Cactophilic and non-cactophilic yeasts can be predicted from genomic and metabolic data with good accuracy.**
(**A**) (left) Distribution of correct random forest (RF) classifications of cactophilic yeasts across the Saccharomycotina
phylogeny. (right) Confusion matrix showing the average number of true positives (37.40), true negatives (39.95), false
positives (14.05), and false negatives (16.60) across strictly cactophilic and transient species resulting from 20 independent RF
runs. On the bottom, the top 10 most important metabolic (presence or absence of growth; in blue) and genomic (presence

or absence of KEGG; in gray) features for the RF classifier ranked according to their importance scores are shown. (**B**) Confusion matrix and top 10 most important features for the RF classifier using only strictly cactophilic species. (**C**) Distribution of the proportion of the most important features to predict cactophily (strict cactophilic and transient) and non-cactophilic species for the entire dataset of 1,154 yeasts. (**D**) Distribution of the proportion of metabolic traits (representing presence or absence of growth under the conditions shown) [51] across cactophilic (strict and transient) and non-cactophilic species. Only metabolic traits for which no more than 50% of data were missing were considered. Additionally, only metabolic traits exhibiting more than 10% prevalence in one of the groups (cactophilic or non-cactophilic) are shown. Statistically significant differences (chi-squared test; * $p$-value < 0.05, ** $p$-value < 0.01, *** $p$-value < 0.001) between the proportions of cactophilic and non-cactophilic species able to grow in each carbon/nitrogen source are shown. The data underlying this Figure can be found in https://doi.org/10.6084/m9.figshare.24114381.

*Phaffomyces* genus (highlighted in light blue, Fig 3A). These results suggest that the phylogenetic relatedness between some cactophilic species did not significantly interfere with the accuracy of the RF classifier.

We next examined the top features that significantly contributed to the RF classifier. The feature with the highest relative importance was growth at 37°C, when analyzing either all cactophilic species (Fig 3A) or just strictly cactophilic species (Fig 3B). Ability to grow at high temperatures was previously found to be prevalent in cactus-associated yeasts [38, 65] and is also an adaptive feature of cactophilic *Drosophila* [48,66]. Growth at 40°C and 42°C were also among the top 10 most important features, supporting the hypothesis that thermotolerance is a distinctive feature of cactus-associated yeasts. In fact, 90%, 66%, and 46% of cactophilic yeasts can grow at 37°C, 40°C, and 42°C, respectively, compared to only 39%, 19%, and 10% of non-cactophilic yeasts (chi-squared test, $p < 0.01$) (Fig 3C).

Analyzing the top 100 most important metabolic features, we observed that 74 are less common in cactophilic species compared to non-cactophilic species (S2 Table). For instance, trehalose assimilation is more rarely found in cactophilic species (approximately 25%) than non-cactophilic species (approximately 74%) (chi-squared test, $p$-value < 0.01) (Fig 3C). Trehalose generally accumulates during numerous stress conditions including heat stress [67,68]. When cells return to a more favorable condition, the accumulated trehalose is hydrolysed into glucose by the trehalase Nth1. Inactivation of *NTH1* by mutations, resulting in impairment of trehalose hydrolysis, can be one of the outcomes of the heat stress response in experimentally evolved strains of *S. cerevisiae* under high temperature stress [69,70], suggesting that deficient trehalose hydrolysis can be beneficial under long-term thermal stress conditions. We found that *NTH1* is generally present in cactophilic genomes, suggesting that absence of *NTH1* does not explain impairment in trehalose assimilation in these species. Other top important metabolic features, such as sucrose assimilation, are also more rarely found in cactophilic species (chi-squared test, $p$-value < 0.001) (S2 Table and Fig 3C), echoing the general trend of a narrower spectrum of carbon sources assimilated by cactophilic species compared to their non-cactophilic counterparts (Fig 3D). However, some metabolic traits are more frequently found among cactus-associated yeasts, such as assimilation of lactate (chi-squared test, $p$-value < 0.01) (Fig 3D), which was previously found to be positively associated with yeast species isolated from the cacti environment [38,65].

Among the most important genomic features were presence or absence of genes involved in multiple distinct functions: K15325 (splicing), K06116 (glycerol metabolism), K01192 (N-glycan metabolism), K00547 (amino acid metabolism), K11762 (chromatin remodeling), K11370 (DNA repair), K11511 (DNA repair), or K19783 (postreplication repair). All these features are less common in cactophilic than in non-cactophilic species (S2 Table). One interesting exception is K03686, an Hsp40 family protein encoded by 85% of cactophilic and 57% of non-cactophilic species (S2 Table). To ascertain whether metabolic, genomic, or both types of data were driving the classification, we next ran RF using each of the 2 types of data (genomic

and metabolic) separately (S3 Fig). Accuracy values for predicting strictly cactophilic species using only genomic data, only metabolic data, or both were similar (72.2%, 76.4%, and 76.0%, respectively). Importantly, we found that the top features in the RF classifier strongly overlap between the analyses of the different features separately and together (S2 Table), indicating that both metabolic and genomic features are robust fingerprints of adaptation to the cacti environment.

## HGT and duplication of cell wall–degrading enzymes in cactophilic yeasts

We next looked for genes that might be implicated in cactophily by examining gene duplication and HGT across 3 groups that contained 2 or more cactophilic lineages. We constructed 3 distinct datasets (S3 Table) containing multiple cactophilic species and closest non-cactophilic relatives within the Lipomycetales/Dipodascales/Trigonopsidales orders (referred to as LDT group, including *Tortispora* spp., *Dipodascus australiensis*, *Magnusiomyces starmeri*, *Myxozyma mucilagina*, and *Myxozyma neglecta*), Phaffomycetales (including *Starmera* spp. and *Phaffomyces* spp.), and Pichiales (including 2 distinct *Pichia* spp. cactophilic clades).

We focused on gene duplications and HGT, as gene losses are usually not reliably estimated due to annotation and sampling issues or inaccurate gene family clustering [71]. By inspecting orthogroups that uniquely contained cactophilic species, we found a gene encoding a pectate/pectin lyase was uniquely found in *P. eremophila* (strictly cactophilic) and *P. kluyveri* (transient), which are commonly isolated from rotting cacti tissues. Sequence similarity searches across the entire dataset of 1,154 yeast genomes confirmed that this gene is absent from all other species. Pectate lyases are extracellular enzymes involved in pectin hydrolysis and plant cell wall degradation. Consistent with this function, these enzymes are mostly found among plant pathogens and plant-associated fungi and bacteria [72,73], and their activity has only been reported in a handful of Saccharomycotina species [74]. Phylogenetic analyses showed that the 2 yeast sequences are nested deeply within a clade of bacterial pectin lyases (Fig 4A). The most closely related sequence belongs to *Acinetobacter boissieri* [75], which has been frequently isolated from plants and flowers, and to *Xanthomonas* and *Dickeya*, 2 genera of plant pathogenic bacteria [76,77].

Using gene tree–species tree reconciliation analyses implemented in GeneRax [71], we next examined genes with evidence of duplication in at least 1 cactophilic species belonging to each group, while excluding events of duplication in non-cactophilic species belonging to each of the 3 lineages/groups inspected (S4 Table).

Duplication of another gene involved in plant cell wall degradation, encoding a rhamnogalacturonan endolyase (K18195), was detected in the cactophilic *P. antillensis*, *P. opuntiae*, and *Candida coquimbonensis* (Phaffomycetales; S4 Table). These species contained 2 copies of this gene compared to their closest relatives, which contained only one (Fig 4B). These enzymes are responsible for the extracellular cleavage of pectin [82], which, along with cellulose, is one of the major components of plant cell wall. Pectin lyase [74] activities have been rarely reported among yeasts; therefore, we assessed the distribution of the rhamnogalacturonan endolyase across the 1,154 proteomes using a BLASTp search (e-value cutoff $e^{-3}$) and found that it displayed a patchy distribution, being found in fewer than 60 species (Fig 4B).

Consistent with their function, HGT-derived pectin lyases and rhamnogalacturonan endolyases were predicted to localize to the extracellular space based on primary sequence analyses [78] (Fig 4). Pectin lyase enzymatic activity was previously detected in *P. kluyveri* strains associated with coffee fermentation [73,83], suggesting that the identified HGT-derived pectin lyase is likely responsible for this activity.

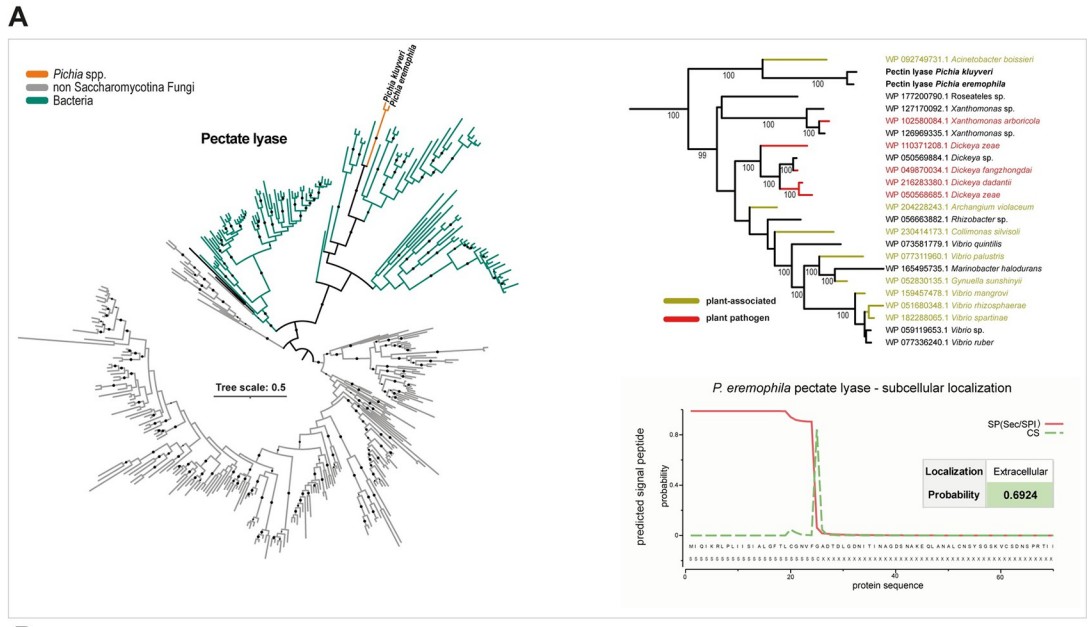

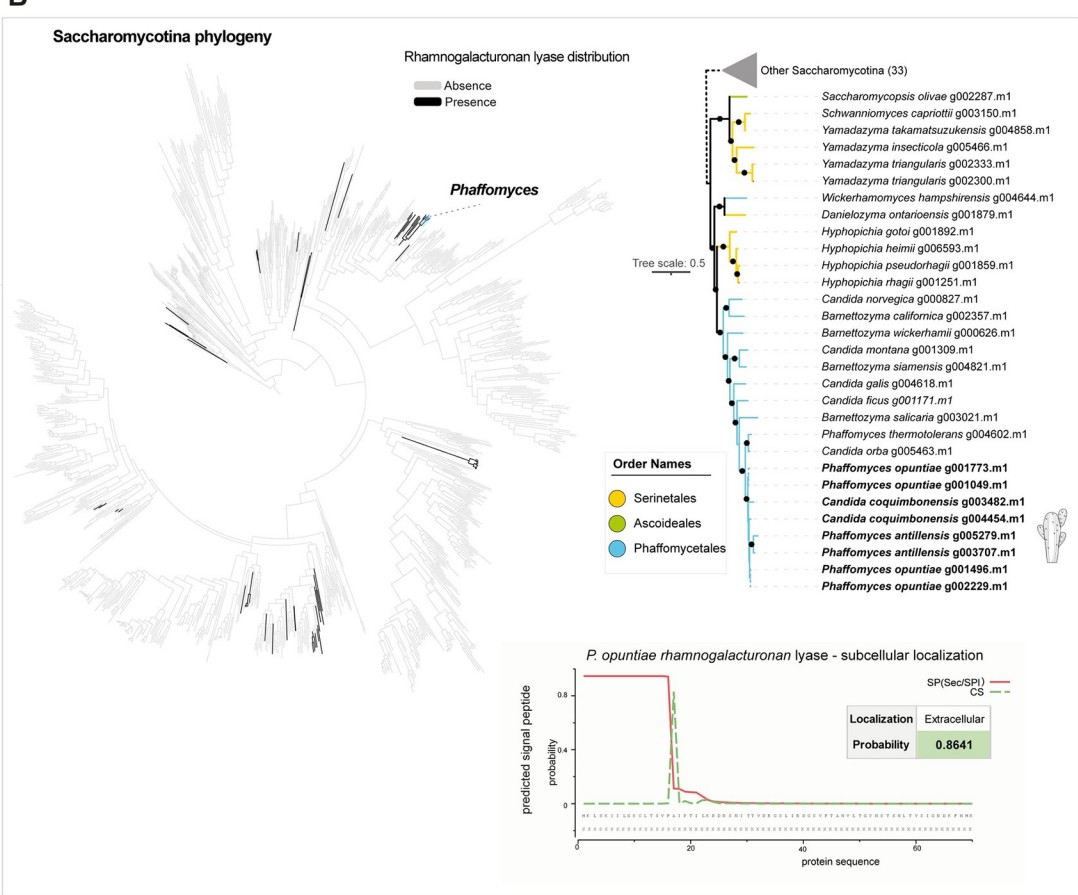

**Fig 4. Duplication and horizontal gene transfer of plant cell wall-degrading enzymes in cactophilic species.** (**A**) Phylogenetic tree of closest related sequences to pectin lyases from *P. eremophila* and *P. kluyveri*. (on the right) Pruned tree highlighting the ecological association of bacteria species harboring the closest related pectin lyase sequences to *P. eremophila*/*P. kluyveri* proteins. Prediction of subcellular localization [78,79] is shown in the panel below. (**B**) Distribution of rhamnogalacturonan endolyases across the 1,154 yeast genomes (presence in black and absence in gray). (on the right) Pruned phylogenetic tree of yeast

rhamnogalacturonan endolyase and closest relatives, highlighting the duplication events in cactophilic *Phaffomyces* species. A BLASTp search against the yeast dataset of 1,154 proteomes was performed, and all significant hits were retrieved (e-value cutoff $e^{-3}$). Phylogeny was constructed in IQ-TREE v2.0.6 (-m TEST, -bb 1,000) [80,81]. Branch support (bootstrap > = 90) is represented as black circles. Branches are colored according to taxonomy as indicated in the key. Branches clustering other putative rhamnogalacturonan endolyase sequences in the Saccharomycotina (total of 33) were collapsed. In *P. opuntiae*, there are 2 additional partial sequences (g001496.m1 and g001773.m1, 163 amino acids) that only partially overlap (from 39 to 163 overlapping amino acids) with the remaining nearly complete sequences (g002229.m1: 468 amino acids and g001049.m1: 610 amino acids). Prediction of subcellular localization according to SignalP and Deeploc [78,79] is shown in the panel below. The data underlying this Figure can be found in https://doi.org/10.6084/m9.figshare.24114381.

## Convergent accelerated rates in heat resistance-related genes

We next specifically looked for evidence of convergent evolutionary rates [64], another indicator of adaptation [21,22]. For this analysis, we inspected the selected the same lineages/groups as for the gene family analyses (Pichiales, Phaffomycetales and the LDT group; S3 Table).

Correlation analyses between relative evolutionary rates (RERs—the rate at which a given branch on a gene tree is evolving, normalized by the genome-wide evolutionary rate on that branch) and cactophily across ancestral and terminal branches revealed changes in evolutionary rate associated with the evolution of cactophily. Specifically, we inferred that 20/3,029 gene families in the LDT group are under accelerated evolution (i.e., these genes are evolving significantly faster in cactophilic lineages than in their non-cactophilic relatives) and 33/3,029 have undergone decelerated evolution (i.e., these genes are evolving significantly slower in cactophilic lineages than in their non-cactophilic relatives) (S5 Table and Fig 5A). In the Pichiales, 14/2,204 showed evidence of acceleration and 25/2,204 of deceleration (Fig 5A). In the Phaffomycetales, we found 32/3,550 accelerated genes and 30/3,550 decelerated genes (S5 Table and Fig 5A).

The accelerated genes are associated with varied cellular functions; however, no significant enrichment in particular biological or molecular functions was found (S5 Table). In Pichiales, 7 out of 14 accelerated genes impact heat resistance according to large-scale studies in *S. cerevisiae* [84]. For instance, *SWI6* encodes a transcription factor that induces transcription during heat stress, and deletion of this gene causes several impairments in the resistance to multiple stresses, including heat [85] and cold [86]. Among the accelerated genes in Phaffomycetales, we found *LEC1*, which was recently associated with ergosterol organization (Fig 5B) [87].

Inspecting the literature for phenotypes associated with either null or conditional mutants in *S. cerevisiae* [84], we found that, irrespective of their function, 12/20 genes that exhibited accelerated rates in the LDT group are involved in either heat and/or desiccation resistance. For instance, loss-of-function mutations in *CAP2*, which encodes part of a capping complex involved in barbed-end actin filament capping and filamentous growth, are associated with heat sensitivity and abnormal chitin localization leading to aberrant cell morphology [88,89]. Another gene identified as having accelerated evolutionary rates in the LDT group was *PHO23* (Fig 5B), which was found to be required for the growth of *S. cerevisiae* during heat shock [85]. Genes that underwent decelerated evolution also play multiple roles (S5 Table), and some are involved in essential functions such as DNA repair, cell cycle, and splicing or encoding ribosomal proteins.

We next assessed the occurrence of positive selection across cactophilic clades using branch-site tests of rates of nonsynonymous (dN) and synonymous substitutions (dS) [90,91]. These tests were performed separately for each of the 5 cactophilic subclades within the major lineages selected: *Pichia* A and *Pichia* B clades (Pichiales), *Starmera* and *Phaffomyces* clades (Phaffomycetales), and *Tortispora* clade (Trigonopsidales–LDT group) (S4 Fig). First, we ran CODEML [92] on foreground branches (*p*-value < 0.005, Benjamini–Hochberg (BH)

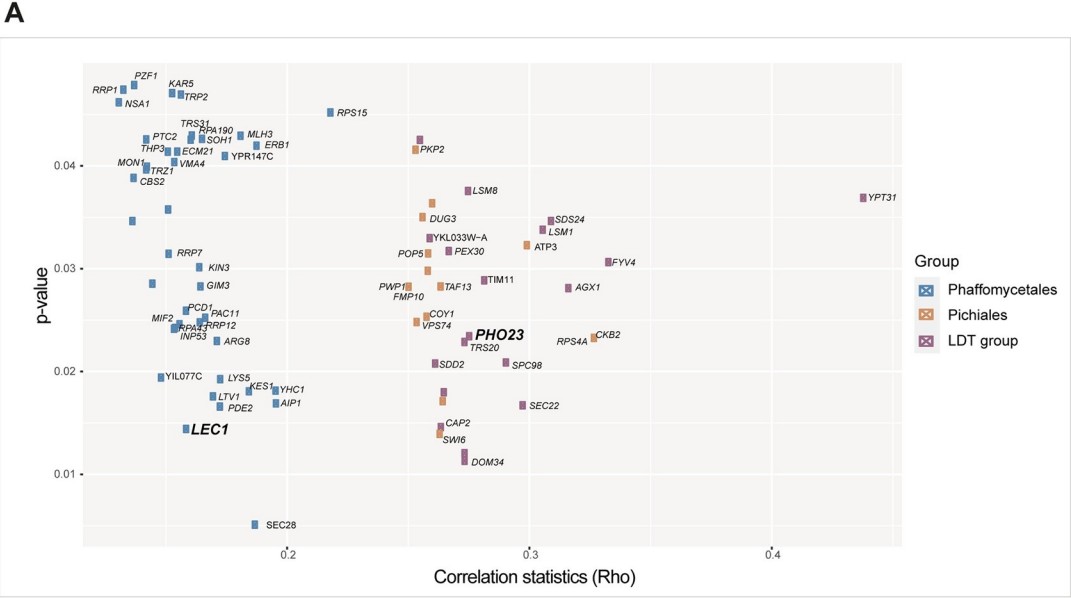

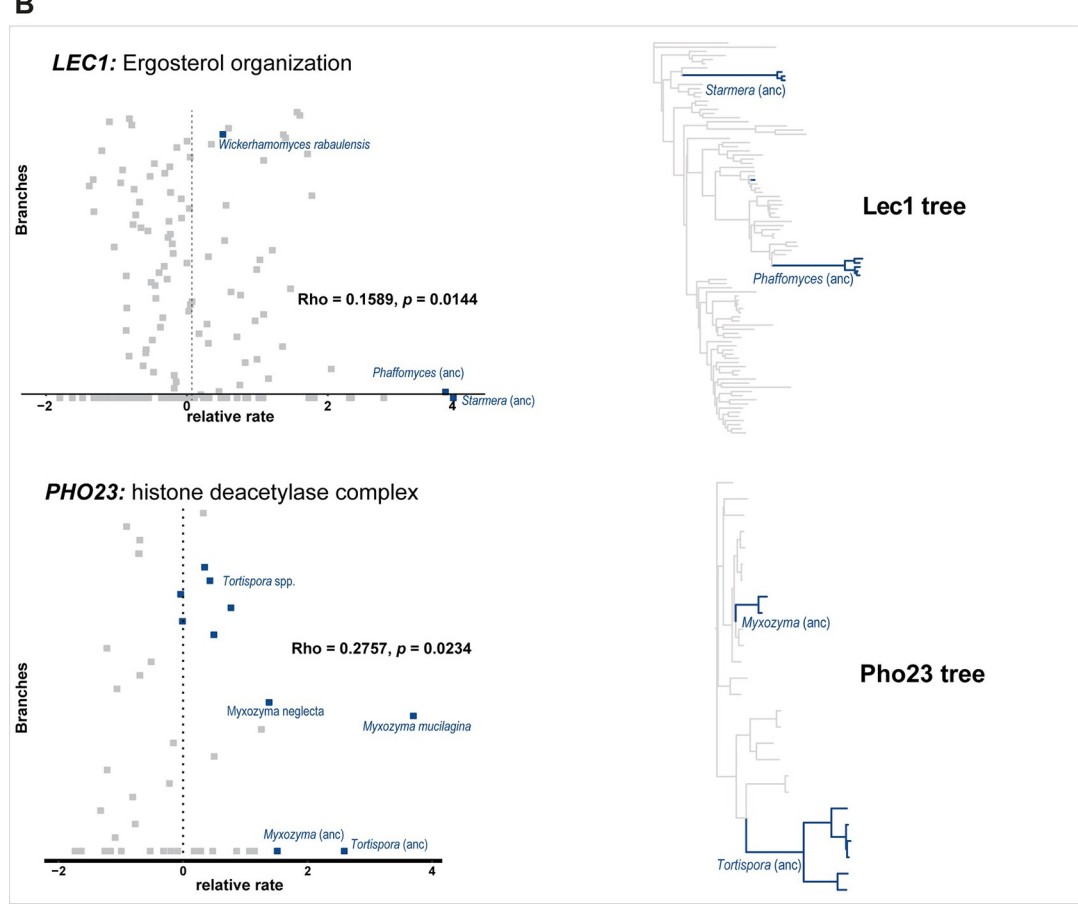

**Fig 5. Genes involved in maintenance of the cell envelope show accelerated evolution in cactophilic clades.** (**A**) Genes with accelerated evolutionary rates in the 3 cactophilic groups inspected. Correlation statistics (Rho) for convergence between accelerated evolutionary rates and cactophily, as well as the respective statistical significance (*p*-value) of this correlation. (**B**) (left) Relative evolutionary rates (RER) and (right) fixed-topology trees for proteins related with integrity of the cell membrane (Lec1) and heat stress (Pho23). The data underlying this Figure can be found in https://doi.org/10.6084/m9.figshare.24114381.

correction) and discarded genes for which evidence of positive selection, even if only marginal, was found in the non-cactophilic sister clades (*p*-value < 0.05, no correction) (S4 Fig). We then validated genes for which a strong signal for selection was found specifically in cactophilic taxa but not in their non-cactophilic relatives with aBSREL [91] (*p*-value < 0.05, no correction). With this conservative approach, we found evidence for positive selection for 38/2,279 genes examined in *Tortispora*; 285/2,175 in *Pichia* A; 112/2,155 in *Pichia* B; 99/1,685 in *Phaffomyces*; and 68/1,510 in *Starmera* (S6 Table). Importantly, signatures of selection (significantly higher ω values than 1 in branches leading to cactophilic clades) can stem from either high dN or low dS values [92], and we did not specifically distinguish between the 2 scenarios. The strongest candidate genes to be under positive selection showed limited overlap; 59 genes in 2 or more clades, while no genes presented evidence for positive selection in all 5 clades.

The 59 genes with evidence of positive selection in 2 or 3 clades are involved in multiple biological functions (S6 and S7 Tables). We noted that several genes involved in ergosterol biosynthesis were under positive selection in multiple cactophilic clades (*ERG1*, *ERG13*, *ERG24*, *ERG26*, *UPC2*, *SIP3*, *HMG1*, and *LAF1*) (Fig 6 and S6 Table). Part of these genes are under positive selection in more than 1 clade (*ERG1*, *ERG26*, and *UPC2*). Ergosterol is involved in stabilizing cell membranes during heat stress and therefore has a major role in the tolerance to numerous stresses in fungi [68,93–95].

Other genes were specifically involved in cell wall biosynthesis and integrity (S6 Table and Fig 6). For instance, *CDA2*, which encodes a chitin deacetylase involved in the function of the fungal cell wall [96,97], showed evidence of positive selection in the stem branches of 3 distinct

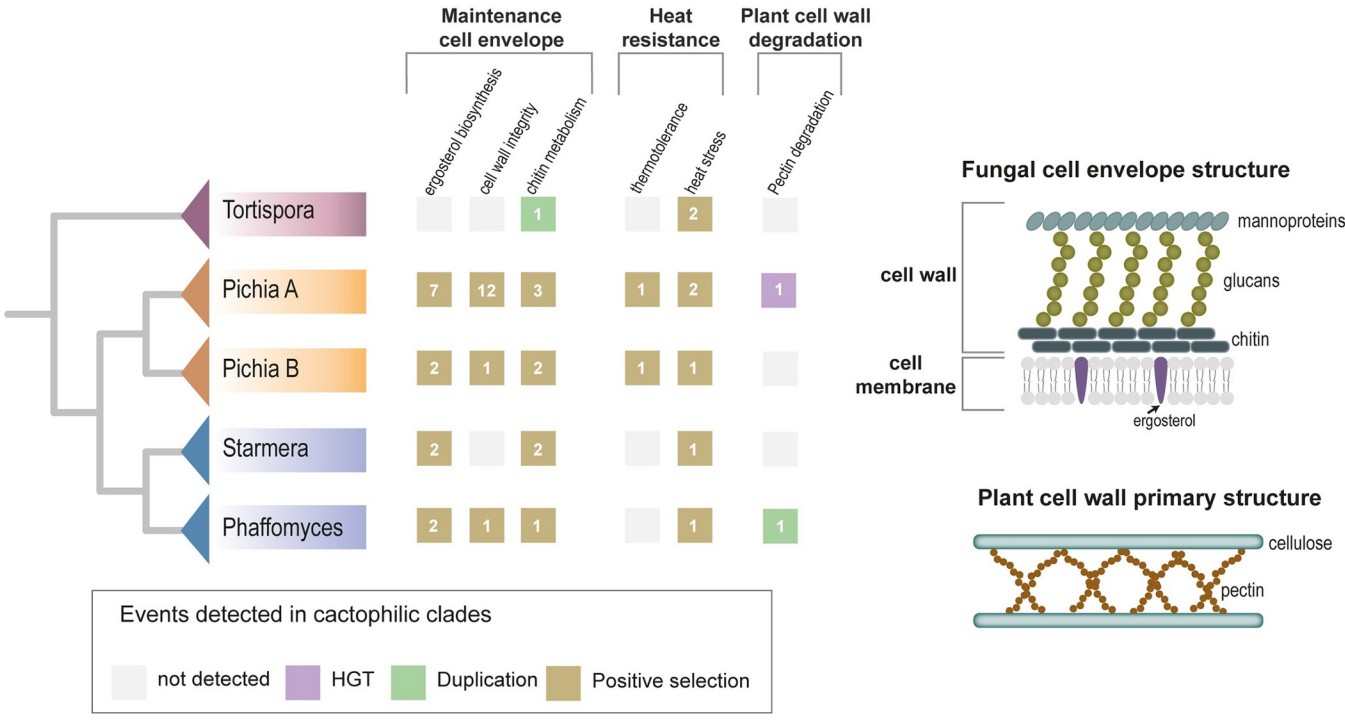

**Fig 6. Signatures of convergent molecular evolution across cactophilic clades.** General biological functions/pathways for which detection of distinctive evolutionary alterations (positive selection, duplication, and HGT) in cactophilic clades are highlighted. The distinct evolutionary events are represented by squares filled with different colors, as indicated in the key. The number of genes involved in each function/pathway is indicated in the squares (identification of these genes can be accessed in S6 Table). Schematic representations of the fungal cell envelope and plant cell wall are shown as many of the functions with signatures of convergence impact these 2 structures.

cactophilic clades (*Pichia* A, *Pichia* B, and *Starmera*). In *Tortispora*, we found 2 copies of this gene.

Other cell wall–related functions were found among the group of genes that showed evidence for positive selection in cactophilic clades, namely, chitin synthases *CHS2* and *CHS3* and chitin-related genes *CHS5* (involved in Chs3 transport from the Golgi to the plasma membrane), *UTR2* (encoding a chitin transglycosylase), or *BUD7* (encoding a Chs5 binding protein) [98]. *DCW1* (encoding a mannosidase required for cell wall formation and resistance to high temperatures) [99], *STE7* (encoding a signal transducing MAP kinase involved in cell wall integrity and pseudohyphal growth) [100], and *AYR1* (encoding a bifunctional triacylglycerol lipase involved in cell wall biosynthesis) [101] were also found among the strongest candidates. Mutations in these genes have been associated with cell wall defects and increased heat sensitivity in *S. cerevisiae* [102–105], and differential transcriptional responses to heat stress have also been documented for some of these genes and functions [106].

While we previously found that the distribution of presence/absence of the trehalase gene *NTH1* was not associated with the low prevalence of trehalose assimilation in cactophilic species (Fig 3C), *NTH1* was among the candidate genes to be under positive selection in the 2 *Pichia* clades (S6 Table). We also found several genes involved in the response to heat stress across cactophilic clades (*SSC1*, *HSC82*, *HSP60*, *DBP5*, *AIM10*, *MCK1*, and *NAT1*).

## Genes involved in maintenance of the cell envelope show evidence of codon optimization

To infer the transcriptional activity of cactophilic yeasts, we determined gene-wise relative synonymous codon usage (gw-RSCU), a metric that measures biases in codon usage that have been shown to be associated with expression level [107], and examined the top-ranked genes (95th percentile) in cactophilic species (S8 Table). Top-ranked genes include many encoding ribosomal proteins and histones, which are known to be highly expressed and codon-optimized in *S. cerevisiae* [108]. We noticed that the chitin deacetylase gene *CDA2* and genes involved in ergosterol biosynthesis (namely, *ERG2*, *ERG5*, *ERG6*, and *ERG11*) were among the genes that fell within the 95th percentile rank for gw-RSCU in multiple cactophilic species. To ascertain whether these genes also show signatures of codon optimization in closely related non-cactophilic species, we determined their respective gw-RSCU percentile ranks. While no clear pattern was observed for *ERG* genes (these genes were also highly ranked for gw-RSCU in non-cactophilic species), we observed that *CDA2* is particularly highly ranked in *Phaffomyces*, *Starmera*, and *Pichia* clades compared to their closest relative non-cactophilic species (S5 Fig). *CDA2* also showed evidence for positive selection in both *Pichia* clades and *Starmera*, suggesting that distinctive synonymous and/or nonsynonymous might have resulted from translational selection for optimized codons due to higher gene expression.

## Convergent evolution of yeast cactophily occurred via the independent acquisition of the same phenotypic traits through mostly distinct genetic changes (Scenario II)

Our results suggest that the evolution of cactophily across the Saccharomycotina are most consistent with Scenario II (phenotypic convergence associated with distinct genetic underpinnings; Fig 2). The intermediate accuracy in predicting cactophily obtained with the RF classifier, using both genomic and metabolic features together or separately, and the general overlap between the top features identified across the distinct RF runs, suggests that only some features (both genomic and metabolic) are common to all species, while others may be species-specific. It may also suggest that some common phenotypic features to most cactophilic

species (such as thermotolerance) might have evolved through distinct genetic trajectories (Scenario II). This was further supported by the results obtained from the evolutionary rates analyses where we found mostly distinct genes involved in similar functions (for instance, heat stress response, ergosterol biosynthesis) under selection or accelerated evolution across cactophilic clades. Nevertheless, some exceptions where the same genes were involved (for instance, *CDA2*), offering support for Scenario I, did exist. Finally, we found distinct mechanisms (HGT and gene duplication) acting on distinct genes (encoding pectin degrading enzymes) involved in similar functions (plant cell wall degradation); these findings were also most consistent with Scenario II.

## Cactophily as a launching pad for the emergence of opportunistic human pathogens?

Thermotolerance is a key shared trait by human fungal pathogens [27,109–111]. Interestingly, several cactophilic or closely related species are emerging human opportunistic pathogens (Figs 7 and S6). Examples include *Candida inconspicua* and *Pichia norvegensis* [112], which cluster within the *Pichia cactophila* clade (Fig 7), and *Pichia cactophila*, which was also isolated from human tissue [113]. Recently, a novel and extremely thermotolerant clinical isolate was identified as belonging to the *P. cactophila* clade [114]. Cases of fungemia have also been associated with *Pichia kluyveri* [115], a transient species belonging to a separate clade within the Pichiales. *Kodamaea ohmeri*, which has also been isolated from the cacti environment [52] and is closely related to the cactophilic *Ko. nitidulidarum* and *Ko. restingae*, is also an emerging human pathogen with a significant mortality rate [116–118].

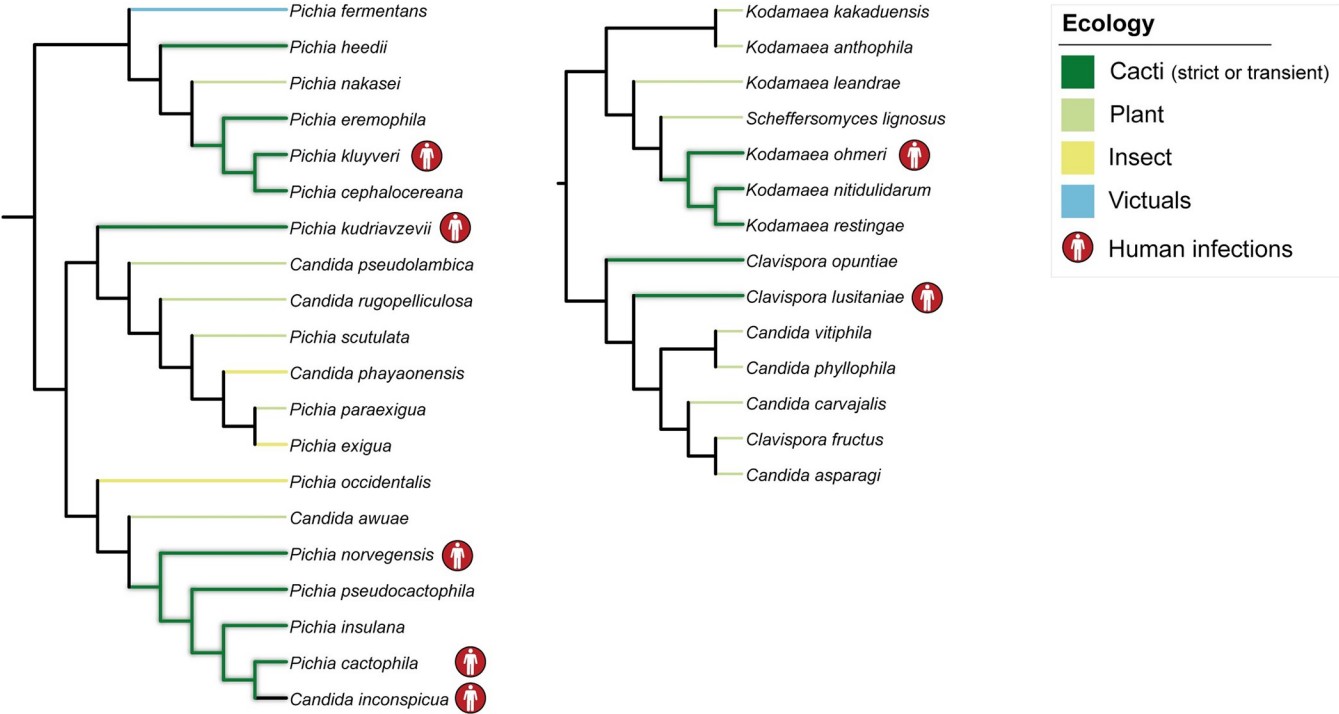

**Fig 7. Example cactophilic lineages that contain human opportunistic pathogens.** Ecological associations of cactophilic species and their closest relatives are represented, highlighting examples of species associated with human infections. Ecological information for additional cactophilic species and closest relatives is provided in S6 Fig.

In addition to thermotolerance, other aspects of the cactophilic lifestyle might be preadaptations for human pathogenicity. For instance, the sterol composition of the cell envelope has been implicated in fungal virulence [95,119–121]. Mutations in genes involved in the ergosterol biosynthetic pathway are associated with antifungal resistance [121–124], making it a major target of antifungal drugs [125]. We showed that the evolutionary rates of several genes involved in the ergosterol biosynthesis across multiple cactophilic clades have changed, suggesting that this pathway might be under a new selection regime in these lineages. However, the impact of these alterations remains to be elucidated.

## Discussion

By examining high-throughput genomic, phenotypic, and ecological data for 1,049 yeast species, we unveiled multiple (approximately 17) independent occurrences of cacti association. The ability to grow at $\geq 37°C$ emerged as the strongest predictor of the cactophilic lifestyle. Being a polygenic trait, thermotolerance can arise through multiple distinct evolutionary trajectories [126]. Heat stress generally affects protein folding and cell integrity and involves a complex response from multiple genes impacting the expression of heat shock proteins, the integrity of the cell wall and membranes, production of compatible solutes, repression of protein biosynthesis, and/or temporary interruption of the cell cycle [67]. The expression of genes involved in cell wall biosynthesis and integrity is, for instance, affected when strains of *S. cerevisiae* are exposed to heat stress [106]. Possibly in association with thermotolerance, we found genes involved in maintaining the cell envelope exhibiting evidence of positive selection, codon optimization, and duplication in multiple cactophilic clades.

We also found genomic fingerprints that indicate phenotypic convergence in the ability to feed on plant material. Acquisition and duplication of plant cell wall–degrading enzymes can be interpreted as adaptive and supports the involvement of cactophilic yeasts in the cacti necrosis process [40]. Interestingly, the contrasting mechanisms employed, HGT of a bacterial pectin lyase and duplication of a rhamnogalacturonan lyase, show that phenotypic convergence can arise through disparate molecular mechanisms. Based on our machine learning, gene family, and evolutionary rates analyses, we detected many cases of phenotypic convergence through distinct molecular mechanisms both in respect to thermotolerance and to plant cell wall degrading ability (Scenario II; Fig 2), as well as a few cases of phenotypic and molecular convergence for thermotolerance (Scenario I; Fig 2). Based on these data, we conclude that cactophily can originate through multiple phenotypic and genetic changes, some commonly found and some more rare, which generally fits best with Scenario II.

It is important to note, however, that cactophily is a complex ecological trait and many genetic and phenotypic changes have likely been involved in its independent evolution across the Saccharomycotina. Some of the phenotypic traits contributing to cactophily may be shared (Scenarios I and II), yet others may be unique to specific taxa (Scenario III). The pleiotropy of many genes and the multigenic character of many phenotypic traits further complicate distinguishing between scenarios (especially Scenarios I and II).

It was previously postulated that most cactophilic yeasts evolved from ancestors associated with plants [37] (S6 Fig), indicating that the ability to thrive in plant-related environments was already present in the ancestors of many of the lineages that evolved cactophilic lifestyle. However, cacti are native to arid and semiarid climates in the Americas [127], where very high (and low) temperatures and low humidity constitute crucial challenges. This reasoning aligns with our finding that thermotolerance is the phenotypic feature with the strongest signature of adaptation to cacti.

Compared to mammals, a lineage that has provided spectacular examples of parallel molecular evolution underpinning the independent emergence of convergent traits [5,8,63,128–130], yeasts exhibit far higher levels of genetic and physiological diversity [26]. Consequently, the probability of finding overlapping evolutionary paths might be reduced as pleiotropic effects or mutational epistasis might be more prominent across divergent genetic backgrounds [9,10]. It is also possible that collection of additional phenotypic data or evolutionary genomic analyses will revise our understanding of the nature of convergent evolution to cactophily. This caveat notwithstanding, these results present a first snapshot of the study of convergent evolution of an ecological trait in yeasts, employing multiple state-of-the-art methodologies that aim at looking into a wide range of evolutionary mechanisms, phenotypes, and genetic determinants. It also underlines the exceptional value of combining high-throughput physiological, genomic, and ecological data to investigate still-pressing questions in evolutionary biology.

## Materials and methods

### Species selection

Yeast species associated were selected from Opulente and colleagues [51] by cross-referencing ecological information available from the literature [33,35–37,39,52,54,55,58–60,62,131,132]. Two different groups of cactophilic species were determined according to their degree of association with cacti: strictly cactophilic (species that are mainly isolated from cacti and very rarely isolated from other environments) or transiently cactophilic (species frequently found in cacti, but also frequently found in other environments or for which strong association with cacti was not clear from the literature) (S1 Table). Species very rarely isolated from this environment were not considered as they could either represent misidentifications or have originated from stochastic events.

### Inference and dating of cacti association events in the Saccharomycotina

The Saccharomycotina phylogeny used throughout this work was inferred in Opulente and colleagues [51]. Briefly, the concatenation-based ML tree was inferred with 1,403 orthologous groups (OGs) of genes using IQ-TREE v2.0.7 [81]. The number of independent events of cacti association were inferred by performing an ancestral state reconstruction using a continuous-time Markov model for discrete trait evolution implemented in Mr Bayes [133]. A simplified workflow implemented in the R environment was followed [134,135]. Only species classified as strictly cactophilic (S1 Table) were considered as exhibiting the trait (cactophily). Transient species were considered as not having the trait. The estimated times for the emergence of cactophily were inferred according to a relaxed molecular clock analyses of the subphylum Saccharomycotina [51].

### Machine learning

To assess whether cactophilic species can be classified based on physiological and/or genomic traits, we used a RF classifier with physiological data (for 893/1,154 strains) and genomic data (functional KEGG annotations) for 1,154 yeast strains obtained from Opulente and colleagues [51]. Briefly, physiological data were generated by assessing quantitative growth for a dataset of 853 strains in 24 different carbon and nitrogen substrates. Functional genomic data (KEGG orthologs) were assigned to protein sequences of gene models using the mapper mode implemented in KofamScan tool [136].

All strictly and transiently cactophilic species identified in the dataset of 1,154 strains were considered for the analyses and were classified as "1" (meaning having the trait). All the remaining species in the dataset were classified as "0" (meaning lacking the trait). The RF classifier is a commonly used machine learning algorithm that is useful for this type of analysis as it can capture interactive effects between features in the training dataset and, and it is straightforward to identify the features that most contribute to the prediction accuracy of the algorithm, facilitating the exploration of large datasets for generation of hypotheses and biological meaning [137]. We trained a machine learning algorithm built by an XGBoost (1.7.3) RF classifier (XGBRFClassifier()) with the parameters "max_depth = 12, n_estimators = 100, use_label_encoder = False, eval_metric = 'mlogloss', n_jobs = 8" on 90% of the data, and we used the remaining 10% for cross-validation, using RepeatedStratifiedKFold from sklearn. model_selection (1.2.1) [137–139]. We used RepeatedStratifiedKFold to generate accuracy measures. We used the cross_val_predict() function from Sci-Kit Learn to generate the confusion matrixes; these matrices show the numbers of strains correctly predicted to be cactophilic or non cactophilic (True Positives and True Negatives, respectively) and incorrectly predicted (False Positives, which are predicted to be cactophilic but are not; and False Negatives, which are not predicted to be, but are cactophilic). Top features were automatically generated by the XGBRFClassifier using Gini importance, which measures a mean decrease in node impurity for each feature, weighted by the probability of each sample reaching that node (nodes are splits on the decision tree and node impurity is the amount of variance in growth on a given carbon source for strains that either have or do not have this trait/feature). This process was repeated for 20 runs using balanced datasets of 54 (or 31 for the analysis excluding transiently cactophilic species) randomly selected non-cactophilic species for each run (and 54 and 31 cactophilic, respectively), and then the averages of each result were used in the final confusion matrixes and feature importance graphs. Balancing helps the RF classifier place as much weight in predicting the rarer, positive cases (cactophilic) as in predicting the more common negative cases (non-cactophilic species). The present dataset is highly unbalanced (less than 5% of the strains have the trait of interest), hence accuracy of predicting cactophily can suffer because predicting its absence in almost all instances leads to very high accuracy, but not very high precision.

## Gene family analyses

To find genes that are specific to or have expanded in cactophilic clades, orthogroup assignment was performed with Orthofinder v.2.3.8 [140] using an inflation parameter of 1.5 and DIAMOND v2.0.13.151 [141] as the sequence aligner. To be able to detect the strongest signatures of adaptation to the cacti environment, we focused only on lineages that contained clades with 3 or more strictly cactophilic species. Due to the high phylogenetic distance between the major strictly cactophilic clades, in order to optimize the number of orthogroups correctly assigned, this analysis was performed separately for each cactophilic group (LDT group, Pichiales, and Phaffomycetales) (S3 Table). Closely related non-cactophilic species were included according to the previously reported phylogeny in Opulente and colleagues [51]. Species in which more than 30% of the genes were multicopy were discarded. Gene family evolutionary history was inferred using GeneRax v1.1.0 [71], which incorporates a Maximum-likelihood and species-tree-aware method. For that, orthogroups containing more than 10 sequences were aligned with MAFFT v7.402 using an iterative refinement method (L-INS-i) [142]. Pruned species trees for each dataset were obtained from the main Saccharomycotina tree [51] using PHYKIT v 1.11.12 [143]. The species trees and alignments were subsequently used as inputs in GeneRax. Briefly, the UndatedDTL probabilistic model was used to compute

the reconciliation likelihood that accounts for duplications, transfers, and losses. For simplification, the same model of sequence evolution was used for all gene families (LG+I+G4) during gene tree inference by GeneRax. Events of duplication occurring in at least 1 cactophilic species were first identified. Subsequently, duplication events occurring in non-cactophilic clades/species within each of the 3 datasets were removed, so as to consider duplication events specific to cactophilic species within each of the 3 lineages/groups analyzed (Pichiales, Phaffomycetales, and the LDT group). Reconciled trees were visualized with Notung v2.9 [144], and phylogenetic trees were constructed for candidate genes/gene families putatively relevant for niche adaptation.

## Evolutionary rates

To determine which genes might exhibit altered evolutionary rates in cactophilic clades/species, we used both branch-site tests of positive selection using CODEML implemented in PAML [90] and absREL implemented in Hyphy [91], and convergent evolutionary rates analyses implemented in RER converge [64]. To investigate fingerprints of convergence in evolutionary rates, we used RER converge using the same groups/lineages as for the gene family analyses (S3 Table) in order to include more than 1 cactophilic clade/species per dataset, so that convergence could be tested. In this case, single-copy orthologs (SCOs) from the Orthofinder run performed for the gene family evolution analysis were used. To increase the number of orthologs available for analysis, multicopy orthogroups that were present in at least 2 species belonging to distinct cactophilic clades within each dataset were selected. Next, SCOs from each multicopy orthogroup were pruned using OrthoSNAP v0.0.1 [145]. To do this, multiple sequence alignments were produced for each multicopy orthogroup using MAFFT v7.402 (—localpair), and phylogenies were obtained with FastTree [146]. We next ran OrthoSNAP with default parameters, keeping at least 50% of the species from the original dataset (50% occupancy). For each orthogroup, branch lengths were estimated on a fixed topology obtained from the Saccharomycotina species tree [51] by pruning the species of interest using PHYKIT v 1.11.12 [143]. Each orthogroup was first aligned with MAFFT v7.402 (—localpair), and the best-fitting model was assessed using IQ-TREE v2.0.6. Branch lengths were determined for each orthogroup, in the fixed tree topology, using RAxML-NG v.0.9.0 [147] under the best protein models inferred by IQ-TREE. All phylogenies were further analyzed with RER converge to find evidence of convergent evolutionary rates in cactophilic species included in each dataset.

Briefly, we tested the hypothesis of convergent evolutionary rates in the ancestral branches leading to the cactophilic clades and/or species. Only phylogenies including a minimum of 2 foreground species and 10 species in total were considered. Genes for which a correlation ratio (Rho- correlation between relative evolutionary rate and trait) higher than 0.25 and a *p*-value (association between relative evolutionary rate and trait) lower than 0.05 were obtained were further considered as good candidates for being under convergent accelerated evolution. For those, original trees were manually checked. For Phaffomycetales, we exceptionally considered rho > 0.15 because we failed to find genes with rho > 0.25.

A detailed scheme of the entire workflow can be found in S7 Fig.

For detecting positive selection, we looked specifically into a narrower phylogenetic spectrum including only individual cactophilic clades, to avoid masking non-convergent signatures of selection (genes that are under positive selection in one cactophilic clade but not in other clades and can therefore not be identified as being under positive selection due to lack of statistical power). We used the same set of lineages/groups as for RER analyses but selected only individual clades containing 3 or more strictly cactophilic. Remaining cactophilic (strictly

or transient) species were excluded. In this way, 5 datasets (see S4 Fig) were considered including members of different lineages (2 subclades within the Phaffomycetales: *Starmera* and *Phaffomyces*; 2 within the Pichiales: *Pichia* A and *Pichia* B; and 1 within Trigonopsidales: *Tortispora*). In *Tortispora*, we also included *T. agaves* because, despite not being associated with Cactaceae species, it is associated with plants with similar characteristics (*Agave* spp.) [33]. For all these 5 datasets, closely related species belonging to each of the families were included based on the species phylogeny based on [51] (S4 Fig). Next, selection of orthogroups using Orthofinder v.2.3.8 was performed for all the species included in the 5 datasets. Clustering of sequences was based on protein sequence similarity and calculated using DIAMOND v2.0.13.151 using an inflation parameter of 1.5. SCOs present in all species in each dataset were selected.

To avoid masking nonparallel signatures of positive selection, each dataset was separately analyzed with CODEML [90] using the branch-site model [148] and considering the branch leading to the cactophilic clade as the foreground branch. The likelihood of a gene being under positive selection was evaluated through a likelihood ratio test [LRT: $2 \times (\ln_{H1} - \ln_{H0})$] [148]. LRT values were subsequently transformed into *p*-values using *pchisq* in R [pchisq(q, df = 1, lower.tail = FALSE)]. These *p*-values were corrected for multiple testing, independently for each clade, using the BH procedure [149] (S9 Table). Genes with a corrected *p*-value lower than 0.005 were retained. We performed a second round of corrections to exclude genes with evidence of positive selection also in closely related non-cactophilic species (background branches). For that, we performed branch-site tests in CODEML in the same way but considering the sister clade as the foreground branch (S4 Fig). Genes for which a marginally significant signal for positive selection was obtained for the non-cactophilic relatives (to be conservative, we used an uncorrected *p*-value < 0.05) were excluded. In this way, we only considered genes for which a strong signal for selection was found specifically in cactophilic taxa but not in their non-cactophilic relatives. We further validated genes inferred (by CODEML) to be under selection specifically in cactophilic taxa with another adaptive branch-site random effects likelihood test (aBSREL), implemented in HYPHY [91]; genes that did not show evidence of selection with aBSREL (*p*-value < 0.05, uncorrected) were discarded.

In summary, genes considered to be candidates under selection specifically in cactophilic taxa in our study had to pass 3 criteria:

i. their <u>BH-corrected</u> *p*-value in the foreground branch was lower than 0.005;

ii. their <u>uncorrected</u> *p*-value in background branches was higher than 0.05; and

iii. their uncorrected *p*-value of foreground branches in aBSREL was lower than 0.05.

### Enrichment analyses

Gene ontology (GO) enrichment analyses were performed for the genes under positive selection in each clade and also for genes that exhibited accelerated or decelerated evolution (RER converge).

First, associated GO terms were obtained for all genes using eggNOG-mapper [150]. For in-clade analyses, enrichment analyses were performed using *P. cactophila* (for Pichia A and PichiaB datasets—positive selection analyses, and the Pichiales dataset–RER converge analyses), *Starmera amethionina* (for *Starmera* dataset–positive selection analyses), *Phaffomyces opuntiae* (for *Phaffomyces* dataset–positive selection analyses, and Phaffomycetales- RER converge analyses), and *T. caseinolytica* (for *Tortispora* dataset–positive selection analyses, and LDT group–RER converge abalyses) whole genome annotations as the background. For the

analyses involving genes under positive selection in 2 or more clades, *P. cactophila* genome annotations were used as the background. GO enrichment analyses were performed using the R package topGO 2.28.0 [151]. Statistical significance was assessed using Fisher's exact test using the default "weight01" method. Correction for multiple testing was performed the "BH" correction method. The results can be assessed in S5 and S7 Tables.

## Codon usage bias

To examine codon optimization in particular genes of cactophilic species, we calculated the gw-RSCU, implemented in Biokit v0.0.9 [107]. This metric was shown to correlate with the tRNA adaptation index (tAI) [107], which measures the translation efficiency by considering both codon optimization and the intracellular concentration of tRNA molecules [152]. The gw-RSCU was calculated by determining the mean relative synonymous codon usage value for all codons in each gene in the genome based on their genome-wide RSCU values. We ranked the genes with the highest gw-RSCU values (subtracting the standard deviation to the gw-RSCU mean value) and looked at the genes falling into the 95th percentile and above (S8 Table). Next, gene functions that were relevant for the cactophilic lifestyle were selected, and their gw-RSCU values were inspected in non-cactophilic closest related species. Briefly, a local BLASTp was used to find the putative orthologs by considering a protein identity of >40%. The top hit was considered to correspond to the orthologous gene; however, whenever multiple hits with similar protein sequence identity were found, the one with the highest rank was considered. The percentile ranking for *CDA2* was determined using the R package *dplyr* [153].

## Supporting information

**S1 Fig. Distribution of cacti-association across the Saccharomycotina.** The Saccharomycotina species tree phylogeny is represented with the respective branch labels as in Opulente and colleagues [51]. Cacti association is shown in the outer circle next to the respective species/strain (dark green–strictly cactophilic, light green–transient). The data underlying this Figure can be found in https://doi.org/10.6084/m9.figshare.24114381.
(PDF)

**S2 Fig. Ancestral state reconstruction of cacti association across the Saccharomycotina.** The number of independent events of cacti association were inferred by performing an ancestral state reconstruction using the MBASR toolkit [135] and the topology of the species tree presented in Fig 1. State 0 (red) means absence of the trait while State 1 (orange) means presence of the trait and were attributed according to S1 Table (considering presence of the trait only the strictly cactophilic species). For each node in the tree, probability of the trait being present (orange) or absent (red) is represented as a pie chart. In each terminal branch, red or orange circles represent the extant state (used as an input) for each species. Numbers next to each node represent the node identification.
(PDF)

**S3 Fig. Prediction of cactophily using only genomic or metabolic traits. (A)** Confusion matrices showing the average number of true positives, true negatives, false positives, and false negatives across strictly cactophilic and transient species resulting from 20 independent RF runs using either only metabolic or genomic data. On the right, the top 10 most important metabolic and genomic features for the RF classifier ranked according to their importance scores are shown. The features that overlap with the top 10 most important features resulting from the RF runs with both genomic and metabolic data (Fig 3) are highlighted in green. The

data underlying this Figure can be found in S2 Table.
(PDF)

**S4 Fig. Positive selection datasets showing the species trees used and the control tests in sister clades.** Branches selected for the analyses are shown (ω) in different colors (green for the foreground/cactophilic lineages and red for the background/non-cactophilic lineages). Only the genes for which no evidence of positive selection on the background lineages were considered for further analyses (please see Materials and methods section).
(PDF)

**S5 Fig. *CDA2* gw-RSCU percentile rank in cactophilic species.** Cactophilic species belonging to *Phaffomyces*, *Starmera*, *Pichia*, and *Myxozyma* genera and their respective non-cactophilic closest relatives (identified as "out") were inspected. The data underlying this Figure can be found in https://doi.org/10.6084/m9.figshare.24114381.
(PDF)

**S6 Fig. Ecological associations of cactophilic species and their closest relatives.** Species that have been isolated from clinical contexts, and are emerging opportunistic pathogens, are highlighted. The ecological information presented was obtained from the CBS database and available literature according to the substrate of isolation of the type strain.
(PDF)

**S7 Fig. Workflow of RER converge analyses.** Schematic representation of the workflow for the detection of convergent evolutionary rates from orthogroup assignment and selection to RER converge analyses.
(PDF)

**S1 Table. List of cactophilic species understudy.** Selection of cactus-associated yeasts according to the literature (relevant references are shown). Classification of cactophilic yeasts (strict or transient), geography and substrate of origin is shown for the type strain.
(XLSX)

**S2 Table. Raw machine learning results.** Features are ranked by feature importance (from most important to less important). Proportion of presence in cactophilic and non-cactophilic species are shown for the top 100 most important features for the RF runs using both genomic and metabolic data. For the RF runs using either genomic or metabolic data separately, only feature importances are shown.
(XLSX)

**S3 Table. List of species used for gene family analyses and RER converge analyses.** Cactophilic species considered for both analyses are highlighted in bold.
(XLSX)

**S4 Table. Gene family analyses results.** Duplication events in at least 1 species belonging to yeast cactophilic group inspected were considered. Only proteins for which functional annotations were possible to obtain from EggNOG are shown.
(XLSX)

**S5 Table. RER converge results.** Genes that underwent accelerated and decelerated evolution as well as their putative functions (according to SGD). GO enrichment results are shown.
(XLSX)

**S6 Table. Results of positive selection analyses from branch-site tests using PAML and abs-REL for the 5 cactophilic datasets understudy (*Tortispora*, *Pichia* A, *Pichia* B, *Starmera*,**

and *Phaffomyces*). Putative genes from each orthogroup were identified after a BLASTp search against the nr NCBI database using *Saccharomyces* (taxid:4930) as the reference. When no hit was obtained for *S. cerevisiae*, a BLASTp against the entire nr database was performed instead.
(XLSX)

**S7 Table. GO term enrichment results.** GO terms were obtained for all genes using eggNOG-mapper for genes under positive selection in each clade and for genes under positive selection in 2 or more clades.
(XLSX)

**S8 Table. Top-ranked (95th percentile) genes for gw-RSCU in cactophilic species.** Mean gw-RSCU values were determined with BioKIT [107] and can be accessed in Figshare (https://doi.org/10.6084/m9.figshare.24114381). *CDA2* percentile ranks used for S4 Fig are shown.
(XLSX)

**S9 Table. Likelihood ratio test (LRT) values and *p*-values (before and after BH correction) for CODEML analyses on cactophilic branches for each of the 5 datasets.**
(XLSX)

## Acknowledgments

We thank members of the Rokas Lab, Hittinger Lab, and members of the Y1000+ Project (http://y1000plus.org) for helpful discussions. This work was performed using resources contained within the Advanced Computing Center for research and Education at Vanderbilt University in Nashville, Tennessee.

## Author Contributions

**Conceptualization:** Carla Gonçalves, Antonis Rokas.

**Data curation:** Carla Gonçalves, Marie-Claire Harrison, Dana A. Opulente, Abigail L. LaBella, John F. Wolters, Xiaofan Zhou, Xing-Xing Shen, Marizeth Groenewald.

**Formal analysis:** Carla Gonçalves, Marie-Claire Harrison, Jacob L. Steenwyk.

**Funding acquisition:** Carla Gonçalves, Jacob L. Steenwyk, Chris Todd Hittinger, Antonis Rokas.

**Investigation:** Carla Gonçalves, Jacob L. Steenwyk.

**Methodology:** Carla Gonçalves, Marie-Claire Harrison.

**Project administration:** Carla Gonçalves, Chris Todd Hittinger, Antonis Rokas.

**Resources:** Carla Gonçalves, Dana A. Opulente, Abigail L. LaBella, John F. Wolters, Xiaofan Zhou, Xing-Xing Shen, Marizeth Groenewald, Chris Todd Hittinger, Antonis Rokas.

**Software:** Jacob L. Steenwyk.

**Supervision:** Chris Todd Hittinger, Antonis Rokas.

**Validation:** Carla Gonçalves.

**Visualization:** Carla Gonçalves, Marie-Claire Harrison.

**Writing – original draft:** Carla Gonçalves, Antonis Rokas.

**Writing – review & editing:** Marie-Claire Harrison, Jacob L. Steenwyk, Dana A. Opulente, Abigail L. LaBella, John F. Wolters, Xiaofan Zhou, Xing-Xing Shen, Marizeth Groenewald, Chris Todd Hittinger, Antonis Rokas.

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
