## [Editor Report · Decision Letter 0]

15 Jul 2024

Dear Antonis, 

Thank you for submitting your manuscript entitled "Diverse signatures of convergent evolution in cacti-associated yeasts" for consideration as a Research Article by PLOS Biology.

Your manuscript has now been evaluated by the PLOS Biology editorial staff, as well as by an academic editor with relevant expertise, and I'm writing to let you know that we would like to send your submission out for re-review.

IMPORTANT: The Academic Editor asked me to send this back out to one of the previous reviewers, but can I just check that you have indeed incorporated the changes that you mentioned in your rebuttal? I *think* you have, but I just thought I'd better check upfront.

However, before we can send your manuscript back to the reviewer, we need you to complete your submission by providing the metadata that is required for full assessment. To this end, please login to Editorial Manager where you will find the paper in the 'Submissions Needing Revisions' folder on your homepage. Please click 'Revise Submission' from the Action Links and complete all additional questions in the submission questionnaire.

Once your full submission is complete, your paper will undergo a series of checks in preparation for re-review. After your manuscript has passed the checks it will be sent out for review. To provide the metadata for your submission, please Login to Editorial Manager (https://www.editorialmanager.com/pbiology) within two working days, i.e. by Jul 17 2024 11:59PM.

Kind regards,

Roli

Roland Roberts, PhD

Senior Editor

PLOS Biology

rroberts@plos.org

---

## [Decision Letter · Decision Letter 1]

22 Aug 2024

Dear Antonis,

Thank you for your patience while we considered your revised "portable peer review" manuscript, "Diverse signatures of convergent evolution in cacti-associated yeasts" for publication as a Research Article at PLOS Biology. This revised version of your manuscript has been evaluated by the PLOS Biology editors, the Academic Editor, and by one of the original reviewers from the previous journal.

Based on the reviews and on our Academic Editor's assessment of your revision, we're likely to accept this manuscript for publication, provided you satisfactorily address the remaining points raised by the reviewer and the following data and other policy-related requests.

IMPORTANT - Please attend to the following:

a) Please change your Title to "Diverse signatures of convergent evolution in cactus-associated yeasts" - our thinking is that as one would say "dog-associated" rather than "dogs-associated," by analogy one should say "cactus-associated" rather than "cacti-associated"

b) Please address the minor remaining concern from the reviewer.

c) Please address my Data Policy requests below; specifically, we need you to supply the data underlying Figs 1ABCD, 3ABCD, 4AB, 5AB, S1, 2, S3, S5, either as a supplementary data file or as a permanent DOI’d deposition. I note that you already have an associated Figshare deposition; this seems quite comprehensive, but can you confirm that it's sufficient to reproduce the Figs?

d) Please cite the location of the data clearly in all relevant main and supplementary Figure legends, e.g. “The data underlying this Figure can be found in S1 Data” or “The data underlying this Figure can be found in https://figshare.com/XXXXXXXX"

e) Please make any custom code available, either as a supplementary file or as part of your data deposition.

We expect to receive your revised manuscript within two weeks. 

*Published Peer Review History*

*Press*

Sincerely,

Roli

Roland Roberts, PhD

Senior Editor

rroberts@plos.org

PLOS Biology

DATA POLICY:

Regardless of the method selected, please ensure that you provide the individual numerical values that underlie the summary data displayed in the following figure panels as they are essential for readers to assess your analysis and to reproduce it: Figs 1ABCD, 3ABCD, 4AB, 5AB, S1, 2, S3, S5. NOTE: the numerical data provided should include all replicates AND the way in which the plotted mean and errors were derived (it should not present only the mean/average values).

CODE POLICY

DATA NOT SHOWN?

REVIEWER'S COMMENTS:

Reviewer #1:

[reviewer #3 at Nature Eco Evo]

As stated in my previous reviews of this manuscript, I am positive about this study. I found both the questions and datasets gathered by the authors to answer them innovative and of broad interest. The statistical issue of multiple testing, raised by all reviewers, has been given extra consideration in this revision, which addresses my last concern. 

Minor:

- Line 422, "These tests were performed separately for each of the five cactophilic subclades": does this also imply that the BH correction was done for each clade independently, or was it conducted using all genes of all five clades simultaneously? (Both approaches can be justified; the precise methodology must be stated for clarity.)

---

## [Editor Report · Decision Letter 2]

5 Sep 2024

Dear Dr Rokas,

Thank you for the submission of your revised Research Article "Diverse signatures of convergent evolution in cactus-associated yeasts" for publication in PLOS Biology. On behalf of my colleagues and the Academic Editor, Sophien Kamoun, I am pleased to say that we can in principle accept your manuscript for publication, provided you address any remaining formatting and reporting issues. These will be detailed in an email you should receive within 2-3 business days from our colleagues in the journal operations team; no action is required from you until then. Please note that we will not be able to formally accept your manuscript and schedule it for publication until you have completed any requested changes.

PRESS

Sincerely, 

Melissa

Melissa Vázquez Hernández, PhD

Associate Editor, PLOS Biology

on behalf of

Senior Editor

PLOS Biology

rroberts@plos.org